# Predicting takeover response to silent automated vehicle failures

**Callum Mole**[1], **Jami Pekkanen**[1,2], **William Sheppard**[1], **Tyron Louw**[3], **Richard Romano**[3], **Natasha Merat**[3], **Gustav Markkula**[3], **Richard Wilkie**[1] *

1 School of Psychology, University of Leeds, Leeds, United Kingdom, 2 Cognitive Science, University of Helsinki, Helsinki, Finland, 3 Institute of Transport Studies, University of Leeds, Leeds, United Kingdom

* r.m.wilkie@leeds.ac.uk

**Data Availability Statement:** The raw data, analysis scripts, and experiment code are freely available on the Open Science Framework (https://osf.io/aw8kp/).

**Funding:** RW, CM, JP, WS, NM, RR, and GM were supported by project TRANSITION (EP/P017517/1)

## Abstract

Current and foreseeable automated vehicles are not able to respond appropriately in all circumstances and require human monitoring. An experimental examination of steering automation failure shows that response latency, variability and corrective manoeuvring systematically depend on failure severity and the cognitive load of the driver. The results are formalised into a probabilistic predictive model of response latencies that accounts for failure severity, cognitive load and variability within and between drivers. The model predicts high rates of unsafe outcomes in plausible automation failure scenarios. These findings underline that understanding variability in failure responses is crucial for understanding outcomes in automation failures.

## Introduction

Automated vehicles (AVs) are developing at a rapid pace, but designing a system that can safely respond to all scenarios within existing road infrastructure remains a huge challenge. Consequently, many believe that AVs need to be treated as fallible systems that require a supervisory (human) driver to take over control when the AV is unable to drive safely.

In many cases, the AV will have an understanding of its inherent system limitations. In these situations the AV can give advanced warning of a *planned* transfer of control (i.e a takeover request) to a human driver in a manner that facilitates successful handovers [1]. However, there will also be cases where the AV's ability to drive safely *and* to monitor its performance, is impaired. These scenarios can arise because the system has malfunctioned, reached a limitation it is not aware of, or unintentionally misclassifies or fails to classify an object in (or feature of) the road environment [e.g. the 2016 Tesla crash where the AV failed to identify a truck; [2]. In these cases, the AV may not explicitly notify the driver. In other words, there will be a "silent failure", and it will be up to the supervising driver to detect that the AV has failed and then to respond safely to the conditions. Throughout this manuscript situations where the AV fails without providing *any* explicit alert to the driver will be referred to as *silent* failures (as per [3, 4]). Human detection of these silent failures in automated lane keeping, the resultant steering responses when regaining control, and how distraction affects these behaviours, will be the focus of this manuscript.

funded by EPSRC, UK. The funders had no role in study design, data collection and analysis, decision to publish, or preparation of the manuscript.

**Competing interests:** The authors have declared that no competing interests exist.

Understanding how humans respond to both planned takeover and silent failure conditions will be crucial to setting safety boundaries of AVs. The considerable research examining planned takeover requests allows manufacturers and legislators to design systems and regulations that support safe AVs (for reviews see [3, 5, 6]). However, adequate safety boundaries cannot be established until researchers can predict with confidence how humans respond to silent failures that could, hypothetically, occur at any point during automated driving.

Silent failures will be unpredictable, and it is, therefore, reasonable to expect that their outcomes will likely be more critical than those of planned transfers of control. They will require a driver to act quickly to change the vehicle's motion. To design safe systems, one needs to be able to predict human performance in hypothetical scenarios that vary in criticality (i.e. how much time the driver has to respond before the situation becomes unsafe).

When making predictions based upon hypothetical scenarios, a common approach is to use mechanistic models (i.e. models that describe how perceptual inputs are related to control) to simulate driver behaviour and determine the situations that will be the most problematic. Piccinini et al. [7] have had some success at computationally capturing braking reaction times during silent adaptive cruise control failures. Drivers had longer reaction times than when manually driving, and also longer reaction times for less critical failures. These trends were replicated by extending manual braking models—that accumulate perceptual error signals (e.g. looming; [8]) over time—to automation, by either slowing the rate that perceptual error is accumulated or by incorporating predictions of the AV behaviour into the accumulation process (so 'expected' looming is ignored and not accumulated). Both mechanisms (i.e. prediction and error accumulation) have been suggested to play a role in manual steering corrections [9], but as yet have not been employed to examine the *steering* response to silent failures. Dinparastdjadid et al. [10] showed that a popular model of manual steering control, where drivers generate control outputs based on a weighted combination of angular inputs from a near and a far point [11], can capture the lane position and orientation profiles of steering recoveries to silent failures (where the vehicle drifted without warning while the driver was looking towards a visual distraction task) but crucially fails to describe how the driver moves the steering wheel. Further development is clearly needed for models to capture the mechanisms underpinning steering behaviour in silent failures [3, 10].

The lack of model development is partly due to a lack of empirical work on which to base these models. To the authors' knowledge, there are very few studies that have examined steering responses to automation failures without any alert (exceptions being [10, 12–14]), or with a visual-only alert [which effectively becomes a silent failure when the visual icon is not in the driver's current field of view; e.g. [4, 15]. It appears that under laboratory conditions drivers can respond fairly quickly (in the region of 1-2 s) to silent automation failures when there is a relatively critical and obvious need for a steering intervention [4, 12, 13], though it may take considerably longer for the steering response to stabilise [15].

An important influence on driver responses during planned takeovers and silent failures is the extent to which the driver is engaged in tasks that divert resources from supervising the AV [16]. In silent failure paradigms, reaction times have been reported to be slower when drivers were engaged in additional non-driving-related-tasks that added to the cognitive load [4, 12], which then appeared to propagate through to other metrics of steering, such as increasing maximum steering wheel angles by $\approx$ 15% [12] and leading to more lane excursions [4]. These findings align with some key findings in the literature on planned takeovers, where drivers tend to respond more slowly when cognitively loaded [17–21].

Whilst the previous studies indicate that cognitive load is likely to disrupt driver behaviour during transitions of control, meta-analysis of a wide variety of planned takeover conditions showed that this is not always the case [5]. Cognitive load does generally slow responses, but

when the distraction task is purely auditory (i.e. the task does not need visual attention or a motoric response) there was little difference compared to baseline (non-distracted) conditions [5]. Furthermore, Gold et al. [22] estimated that increased load should increase minimum time-to-collision (i.e. safer responses). The counter-intuitive findings of Gold et al. [22] could be due to drivers overcompensating for potentially delayed responses through more vigorous steering actions when cognitively loaded (cf. increased maximum steering wheel angle in [12]). This explanation has support from research into manual driving (for a review see [23]), in which there have been accounts of cognitive load improving lane keeping (e.g. [24–26]). Yet, there also exist some counter-examples suggesting that cognitive load reduces steering corrections, both in manual driving (e.g. [27, 28]), and also in planned takeovers [29]. The effects of cognitive load on steering behaviour seem to vary depending on the individual and the specific task [23]. In a review of the evidence in manual driving, Engström et al. [23] proposed that cognitive load selectively impacts non-automatised tasks that require cognitive control to enhance weak pathways [30], while keeping well-learned tasks (e.g. lane keeping) unaffected. The influence of cognitive load on steering behaviour during silent failures have not yet been rigorously examined. In the current study, we investigate steering responses under increased cognitive load during highly controlled takeover conditions.

A further factor that influences driver responses is the severity of the failure. In planned failures, drivers take longer to react when the scenario is less critical [19, 31, 32], though the slowing of response does not completely negate the increase in time budget (i.e. drivers respond at a higher time-to-collision for less urgent planned failures [22]. Louw et al. [4]) also found reaction times to be slower, and more variable, for silent failures on straight roads compared to the more critical curves. Greater variability for slower takeover times seems to be a consistent finding across a number of studies [5].

Whilst responses to planned takeovers have often been measured using Reaction Times (RT) there are several limitations to using this metric as a predictor of safety outcomes [33, 34]. Although in most cases an early RT will increase the probability of a safe steering response, RTs cannot be directly mapped onto safe decision-making, or steering (see [6], for a detailed discussion), or braking [33]. The safety relevance of a particular RT can only be realised when placed in context, considering the relationship between the vehicle state (speed, heading, and yaw-rate), road geometry (e.g. road width) when the response is made. Alternatively, one can incorporate the road geometry and the vehicle state *within* the response metric by estimating how long it would take the vehicle to reach the most relevant safety boundary, in the case of driver inaction. For example, some studies use metrics derived from the remaining time until colliding with an obstacle in collision scenarios (e.g. time-to-collision [14, 31, 35]). In a lane keeping scenario (i.e. the current experiment), the relevant metric is time-to-lane-crossing (TLC; [20, 36, 37]). The approach of linking response timings to the relative motion between the vehicle and safety boundaries seems to improve upon RT when predicting safety outcomes, such as crashes when analysing vehicle braking [38] and the rapidity of steering response during AV takeovers [33]. TLC, therefore, is a useful scenario-independent metrics for contextualising the driver's response and will be used here as the key measure of behaviour.

To develop human-centred AV-systems based on drivers' responses to AV failures, it is necessary to consider the *distribution* of responses rather than simply taking mean values [39, 40]. Means can, of course, be useful for establishing average differences between conditions, though this method does aggregate a source of information that is potentially useful for modelling human responses. Using quantile regression, Dinparastdjadid et al. [39] showed that conditions that have a minimal effect on central tendency can have comparatively large effects on the tails of reaction time distributions (during planned takeovers). Furthermore, and more fundamentally, if one is interested in predicting drivers' abilities to respond in real-world

failures, they will need to contend with both between-individual and within-individual variability. Between-individual variability deflects the participant average from the population mean; Within-individual variability causes single responses to failures to be spread around each participant's mean response. Basing predictions on means implicitly aggregates over human variability, yet human variability is an integral component of any real-world failure so arguably should be a key component of applied predictions.

This manuscript provides the first structured examination of human detection and steering response to silent failures. In contrast to previous studies, which examine only a few scenarios (e.g. [4, 10, 12–15]), we systematically examine behaviour across a wide range of failure criticalities in highly controlled takeover conditions. Bayesian hierarchical modelling is employed to closely examine responses to silent failures under both optimal conditions and during increased cognitive load. The stringent modelling captures the between-participant and within-participant variability, leading to applied simulations predicting the safety outcomes of hypothetical real-world failures.

## Results

### Experiment

Silent failures of automation can be classified based upon how quickly the driver would leave the road after the failure in the case of driver inaction ($TLC_F$). The driver is represented by a single point (i.e. a vehicle chassis was not simulated), which is practically similar to calculating TLC from when half the vehicle crosses the lane boundary [20]. Measuring human responses to different criticalities requires several repetitions of the same conditions to gain a reliable estimate of central tendency and variability. Repeatedly presenting only a limited number of failure conditions, however, risks introducing response biases, for example, participants may become highly practiced in responding to a few specific failure types, and the failures themselves become predictable. To counteract this issue, a mixed experimental design was used that combined six repetitions of the same four levels of failure criticality (Repeated) with additional individual trials across a wider range of criticalities (Non-Repeated). See Fig 1 for a graphical description of the failures.

In a driving simulator, participants drove a track consisting of a 2 s straight section connecting to a constant curvature bend of 80 m radius. Trials began in automation, implemented by re-playing the visual scene and wheel movement of a pre-recorded trajectory. Each trial was 15 s long. At a pre-specified amount of time into the trial—the *onset* time—an offset to yaw-rate (i.e. a bias to steering angle) was introduced, so that at each timestep the trajectory's yaw-rate was offset by a constant amount (Fig 1B). In real driving, this type of silent failure might happen for example if the automation is unsuccessful at sensing one of the boundaries of the driver's lane and instead starts following some other marking in the road [41]. After the failure, the yaw-rate no longer matches the road curvature so the vehicle begins to drift towards the road edges (at different rates depending on the severity of the failure; see Materials and methods). The supervising automation task instructions were: "your task as the supervisory driver is to make sure the vehicle stays within the road edges". Manual takeover was achieved by pulling a paddle shifter behind the steering wheel.

An Auditory Continuous Memory Task (ACMT; [42]) was used to introduce cognitive load without visual demand (over-and-above the demands required to complete the steering task). Drivers pressed a button (placed on the front of the wheel) whenever they heard target letters present amongst a stream of distractor items. At the end of each trial they reported how many of each target they thought they had detected Fig 1D & 1E). Throughout the manuscript,

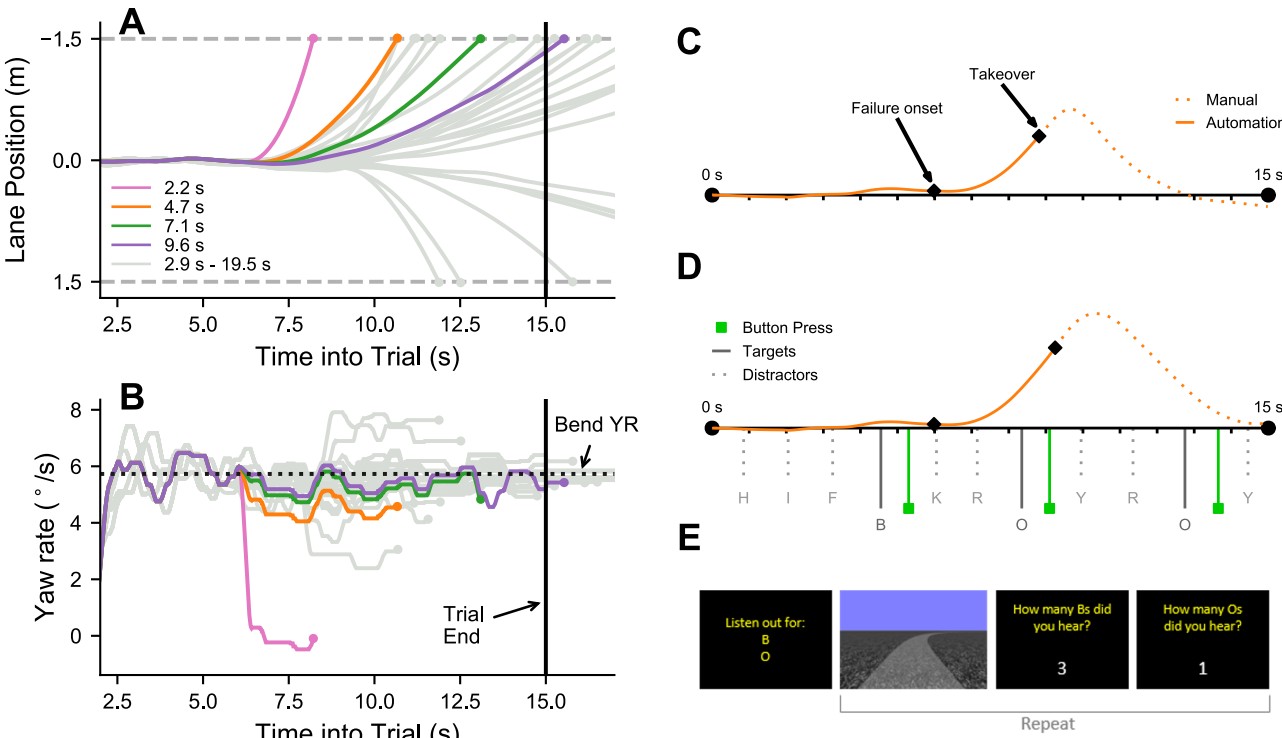

**Fig 1. Simulated failures.** (A) The trajectories of the simulated failures across the entire trial, including the different replayed trajectories (automation), and varied failure onset times. In the figure the road has been straightened out, with the horizontal dashed grey lines indicate the road edges and negative Lane Position values correspond to understeering. (B) The yaw-rate profiles of the simulated failures. Note that the coloured trajectories, which are the Repeated failures (6 repetitions of 4 variations, with an identical replayed trajectory and onset time), all follow the same yaw-rate profile until 6 s, whereupon there is a constant offset to yaw-rate. Solid grey trajectories are the Non-Repeated failures (24 variations), which varied in both replayed trajectories and onset times. Dots correspond to when the trajectory leaves the road. The failure parameter $TLC_F$ is shown in the legend. $TLC_F$ represents the amount of time elapsed between when the failure is introduced and when the driver, represented by a single point, would hit the lane boundaries in the absence of a steering response. Panels C-E show The Trial Sequence. The locomotor component of each trial was 15 s. A bias was introduced in every trial, but severity ranged from negligible to requiring rapid action ([Fig 1]). (C) A sample SupAuto trial (Repeated; $TLC_F$ = 4.7 s; Onset time = 6 s) with the lane position signal overlaid. (D) A sample SupAuto+ACMT trial (Repeated; $TLC_F$ = 4.7 s; Onset time = 6 s) showing the ACMT presentation timings and the participant's button responses. (C) The trial sequence for SupAuto+ACMT. Two target letters were presented at the start of each block of trials. Each trial consisted of the supervising automation task (visual scene shown), followed by the participant estimating how many of each target they had heard. For SupAuto blocks (without the cognitive task), there was a brief blackout at the end of each trial, then the visual scene was reset. For more information see Materials and methods.

the supervising automation task without cognitive load is termed *SupAuto*, and the supervising automation task with the ACMT is termed *SupAuto+ACMT* (see Materials and methods).

## Analytical approach

The analysis presented here uses Bayesian hierarchical models to employ two, complementary, approaches to statistical inference: estimating effect sizes and prediction. The usual inferential approach in experimental psychology is to establish the size or presence of differences between the expected average performance of different conditions (i.e. effects). In hierarchical models, the fixed effect coefficients can be interpreted as the independent contribution of the associated predictor on the population average (i.e. the regression line).

Using a Bayesian approach, each parameter has an associated posterior probability distribution that characterises the level of certainty in parameter values, conditioned on the data. Each parameter's posterior distribution is described using the mean and the 95% highest density interval (HDI), which is the span of the posterior distribution within which there is 95%

probability that the true parameter value will fall, such that values inside the HDI have higher credibility than those outside the HDI [43]. The reader is encouraged against dichotomous thinking of assessing the *presence* of an effect (e.g. by assessing whether a 95% HDI range excludes zero), and asked instead to use the mean and 95% HDIs as estimates of the certainty around the influence of the associated independent variable on the predicted behaviour. Where it is illustrative, we report the percentage of the distribution either side of zero to convey the uncertainty in the model's estimates.

The population average is limited, however, in that it does not contain the within- and between-individual variability that are essential components of any real-world observed takeover. While establishing effects is theoretically useful, population means only exist in an abstract sense and they are a poor model for applied predictions. Bayesian hierarchical models are generative, so predictions of future observations can be made that average over parameter uncertainty [44]. Therefore, throughout the results predictive intervals are reported, which include the variability inherent in any real-world response. These are the intervals that the model believes will encompass individual failures for new (untested) drivers. For predictive intervals, we report the average prediction and intervals for one (68.3%) and two standard deviations (95.5%) away from the mean. Reporting both effect sizes and predictive intervals mean that the practical importance of the results can be robustly assessed.

## Detecting failures: TLC at takeover

In the Introduction, we argue that metrics that are linked to the unfolding scenario should provide better indicators of safe takeover than reaction time, so the measure of detection is time-to-lane-crossing at takeover ($TLC_T$). The timestamp of when the driver pulled the paddle shifter behind the steering wheel was taken as the takeover moment. Note that in the current design the failures are specified in terms of $TLC_F$ so $TLC_T$ can be directly linked to reaction time ($TLC_T = TLC_F$—RT). Trials where the driver takes over control before the failure onsets were removed (2.5% of trials). One participant was removed due to consistently moving the wheel during the period of automation. Of the remaining trials, $TLC_T$ can only be measured in trials where drivers took over control before the trial ended (85.6% of trials). For the less severe combinations of $TLC_F$ and onset time, there is a TLC threshold at the end of the trial, beyond which responses cannot be observed ($TLC_{End}$; Fig 2A).

We found that $TLC_T$ could be reasonably approximated by a normal distribution, with variance increasing as $TLC_F$ increases (Fig 2A). The population mean of $TLC_T$, $\mu$, is modelled as a linear model consisting of an intercept ($\beta_0$), $TLC_F$ (*F* in Eq 2; the corresponding coefficient is denoted $\beta_F$) and Load (*L*; $\beta_L$), including an interaction term ($\beta_{FL}$). Load is parameterised as $L \in \{0, 1\}$, where $L = 1$ means the ACMT is present.

To account for heteroscedasticity, the standard deviation of the response ($\sigma$) is independently modelled in a manner similar to $TLC_T$, with parameters $\alpha_0, \alpha_F, \alpha_L$. Since $\sigma$ cannot be negative, $\ln(\sigma)$ is predicted. To retain a potential for a linear relationship between $TLC_F$ and $\sigma$ (cf. [5]), we log-transform $TLC_F$ when predicting $\sigma$. The resulting model is a multiplicative heteroscedastic model [45].

To exploit the repeated measures design and to capture between-participant variability, these parameters are allowed to vary between participants. For further modelling details see Materials and methods.

Pooled $TLC_T$ for the SupAuto failures are presented in Fig 2A. Drivers performed well at the supervising task, taking over control within the lane boundaries in every instance. Two important characteristics of the data appear obvious: there is a strong linear relationship

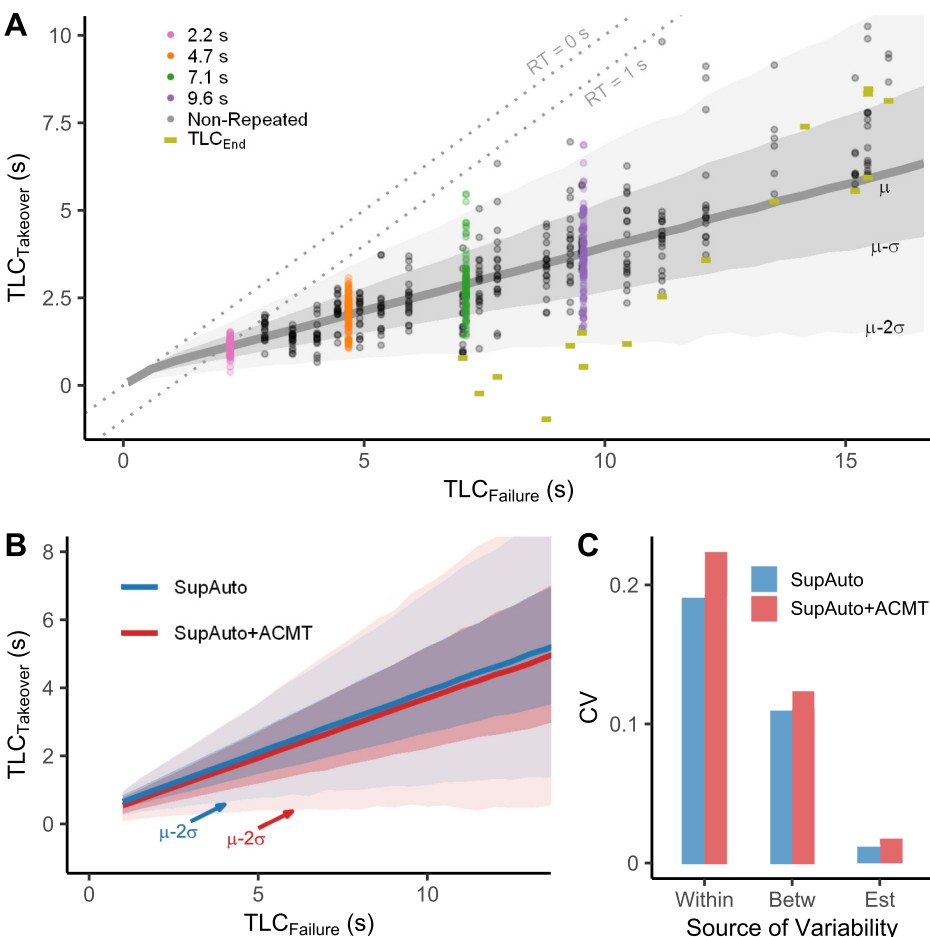

**Fig 2.** (A) Human failure detection data overlaid on model predictive intervals. The pooled data for SupAuto $TLC_T$ is plotted against $TLC_F$. Smaller values of $TLC_F$ indicate more critical failure conditions, whereas, smaller values of $TLC_F$ indicate that the driver took over closer to the lane edge ($TLC_T$). The thick grey solid line is the predicted mean $TLC_T$, with the grey bands showing predictive intervals for one and two standard deviations away from the mean. Coloured dots correspond to the Repeated failure conditions and grey dots correspond to Non-Repeated failure conditions. The $TLC_{End}$ values for each tested combination of $TLC_F$ and onset time, which limits the observed range of $TLC_T$ for the less severe conditions, are shown using gold horizontal bars. To aid interpretation that the reaction times increase as $TLC_F$ increases, two dashed lines with constant reaction times are shown by dashed grey lines (RT = 0s, which is the 1:1 line, and RT = 1 s). (B) Model Predictive Intervals. Regression lines and predictive bounds for 68.3% and 95.5% quantiles for SupAuto and SupAuto+ACMT. (C) The variability within the predictions decomposed into within-participant variability, between-participant variability, and estimation uncertainty, shown as the average contribution to the coefficient of variation ($\sigma_{pred}/\mu_{pred}$) of the predictive distribution. The total (average) coefficient of variation is the sum of the three components. Posterior median parameter values were used to make predictions without estimation uncertainty.

between $TLC_F$ and $TLC_T$ and the variance of $TLC_T$ increases as $TLC_F$ increases. Note that the model regression line and predictive intervals capture the data well.

The coefficient posterior means and 95% HDIs are shown in Table 1. The four $\beta$ parameters predict $\mu$, the mean $TLC_T$. The intercept, $\beta_0$, can be interpreted as the limit of how quickly drivers can respond; the model's estimate is around .33 s. $\beta_F$ predicts how much $TLC_T$ increases for every single unit of $TLC_F$ increase; it is estimated with reasonable certainty to be around .36 s, indicating that 1 s increase in the time budget for a failure translates to approximately .36 s increase in the remaining safety margin when taking over (which, since $TLC_T = TLC_F—RT$ in our setup, means that RTs increased by $\approx$.64 s for every 1 s increase in $TLC_F$). $\beta_L$ corresponds

**Table 1. Posterior means and 95% HDIs for parameters predicting the mean and spread of TLC$_T$, and their estimated variation across the population.**

| Fixed Effects | | | | | Random Effects | | | |
|---|---|---|---|---|---|---|---|---|
| Parameter | Description | Mean | Lower | Upper | $\sigma_{Parameter}$ | Mean | Lower | Upper |
| $\beta_0$ | $\mu$ intercept | .33 | .24 | .42 | $\sigma_{\beta_0}$ | .17 | .1 | .24 |
| $\beta_F$ | TLC$_F$ effect on $\mu$ | .36 | .32 | .41 | $\sigma_{\beta_F}$ | .10 | .07 | .14 |
| $\beta_L$ | ACMT effect on $\mu$ | -.10 | -.19 | -.01 | $\sigma_{\beta_L}$ | .11 | .00 | .19 |
| $\beta_{FL}$ | ACMT × TLC$_F$ effect on $\mu$ | -.01 | -.05 | .03 | $\sigma_{\beta_{FL}}$ | .07 | .04 | .1 |
| $\alpha_0\ [e^{\alpha_0}]$ | ln($\sigma$) intercept [$\sigma$ scaling constant] | -2.47 [.08] | -2.7 [.07] | -2.26 [.11] | $\sigma_{\alpha_0}$ | .33 | .08 | .59 |
| $\alpha_F$ | ln(TLC$_F$) effect on ln($\sigma$) [non-linearity of TLC$_F$ on $\sigma$] | .96 | .84 | 1.09 | $\sigma_{\alpha_F}$ | .21 | .09 | .32 |
| $\alpha_L\ [e^{\alpha_L}]$ | ACMT additive effect on ln($\sigma$) [ACMT scaling effect on $\sigma$] | .09 [1.10] | -.02 [.98] | .24 [1.22] | $\sigma_{\alpha_L}$ | .14 | .0 | .26 |

For $\sigma$, the exponeniated coefficient (that predicts $\sigma$ rather than ln($\sigma$)) is given in square brackets.

to a constant increase or decrease of the regression line when ACMT is present. Though $\beta_L$ is estimated to be small ($\approx$ -.1 s) it is highly likely that ACMT caused a reliable decrease in TLC$_T$ since 98% of the posterior distribution on $\beta_L$ is below zero. $\beta_{FL}$ is estimated, with high certainty, to be close to zero so there is a low likelihood that the presence of ACMT affects the slope of TLC$_T$ to any meaningful degree.

The $\alpha$ parameters in Table 1 predict $\sigma$, the standard deviation of TLC$_T$. An increase in TLC$_F$ increases response variability ($\sigma$). $\alpha_F$ is estimated to be close to one, suggesting that ($\sigma$) increases linearly with TLC$_F$, with a magnitude of approximately 8% of TLC$_F$ magnitude (indicated by $e^{\alpha_0}$ in Table 1). From Table 1 note that there is a high likelihood that drivers' responses were more variable when engaged in the ACMT. Though the mean of $e^{\alpha_L}$ is 1.10 (i.e. ACMT increases $\sigma$ by 10%), the 95% HDIs are relatively wide (-2%—22%; 96% of the posterior > 0) so the magnitude of the proportional increase is uncertain.

One can average over the uncertainty in the posterior distribution when predicting future observations [44]. Fig 2B shows the predicted average mean and predictive intervals for TLC$_T$. In Fig 2B, one can see the lower mean TLC$_T$ and wider predictive intervals for SupAuto +ACMT (cf. parameters $\beta_L$ and $\alpha_L$ in Table 1). However, it is noteworthy that in Fig 2B the predictive intervals are mostly overlapping, and appear large compared to the relatively small effect of ACMT on TLC$_T$.

Since $\sigma$ is explicitly modelled, we can estimate the relative size of different influences on TLC$_T$ bounds when predicting future observations. The predictions contain three sources of variability. Two of these are variability by design: within-participant variability ($\sigma$) and between-participant variability (the varying effects in both $\mu$ and $\sigma$, see Table 1). However, the model also contains estimation uncertainty represented by the posterior distribution of parameters that is taken into account when predicting new observations.

For each condition (a combination of TLC$_F$ and presence of ACMT) there is a predictive distribution, constructed by summing the individual distributions of many simulated drivers (sampled from the random effects based on the structure given in Eqs 4 & 5 and the estimated parameters given in Table 1). To show the relative influences on the spread of this distribution, we use a standardised measure of variability, the coefficient of variation ($CV = \sigma_{pred}/\mu_{pred}$) [46]. Though the CV of the predictive distribution increases slightly over the range of TLC$_F$ owing to the fact that $\sigma$ increases marginally quicker relative to $\mu$, taking the mean CV contribution will suffice for illustrating the relative contributions of within-participant variability, between-participant variability, and estimation uncertainty.

The average CV for the predictive distributions are .3 (SD = .04) for SupAuto and .36 (SD = .05) for SupAuto+ACMT. This means that, on average, without ACMT, the magnitude

of standard deviation is 30% of the magnitude of the mean. The variability breakdown is shown in Fig 2C. The biggest contributor to predictive uncertainty is the within-participant variability (explicitly modelled as $\sigma$), which accounts for around 61% of the total variability. The estimated variability between participants in both $\mu$ and $\sigma$ accounts for approximately 35% of the total TLC$_T$ variability. Between-participant variability is marginally higher for SupAuto+ACMT. The model for SupAuto+ACMT effectively has two additional parameters ($\beta_L$, $\alpha_L$), which each vary between participants (these parameters are zeroed for SupAuto so their variation is omitted from predictions). The additional parameters in SupAuto+ACMT also mean that estimation uncertainty increases (since each parameter brings its estimation uncertainty), but the increase is negligible due to the comparatively small effect estimation uncertainty has on the predictive intervals ($\approx$ 3%).

## Responding to failures: Maximum steering wheel angle

The previous section examined the timing of the immediate response of participants when detecting failure of the automated vehicle. The following analysis examines the nature of the steering produced.

In general, drivers were able to successfully keep the vehicle inside the lane. Across all participants only on 9 occasions (0.25% of trials) did the driver leave the road. However, if one inspects the median trend lines in Fig 3, one can see that drivers ventured slightly closer to the road-edges when performing the ACMT (Lane Position; Fig 3A). When responding to more critical failures, the drivers appeared to turn the wheel more when they were performing the ACMT (Steering Wheel Angle; S3B Fig), yet steering wheel angle traces are similar for more gradual failures (Fig 3B). The previous section showed that drivers were slower to react and achieved a lower safety margin with cognitive load. Further, reaction times positively correlate with both lane position and steering wheel angle (S2 & S3 Figs). Subsequently, one might

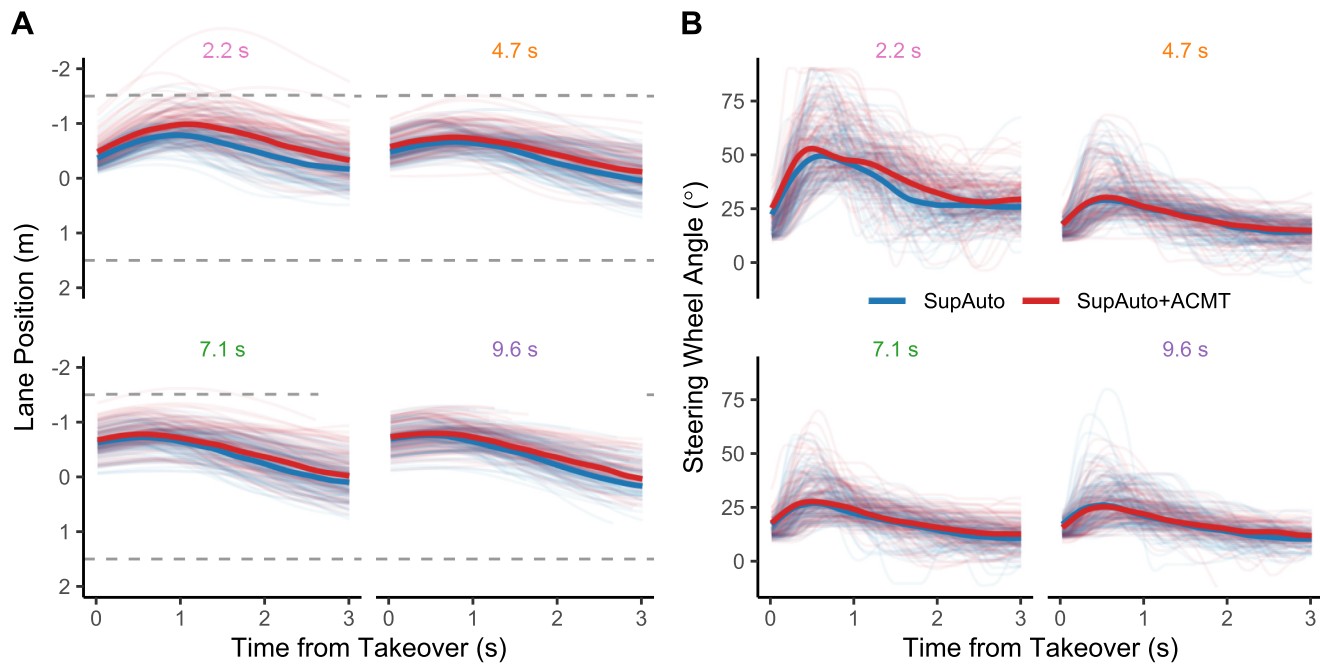

**Fig 3. Steering behaviour.** Individual A) Steering bias and B) Steering Wheel Angle traces for Repeated conditions for the first 3 seconds after takeover, with the rolling average (using a .25 s window) median trend line shown for SupAuto and SupAuto+ACMT shown in bold. Panel titles show TLC$_F$, coloured as per Fig 1.

expect slower reaction times in SupAuto+ACMT to propagate through to differences in steering metrics. This is the case for lane position (S3 Fig), but intriguingly, there do not appear to be clear global differences between SupAuto and SupAuto+ACMT in terms of steering wheel angle (S3 Fig). An interesting question is the extent that steering behaviour is driven by *indirect* effects (e.g. the ACMT delayed RTs leading to greater criticality at takeover that then translates into steering), or *direct* effects (cognitive load directly alters the steering actions).

The steering response characteristically consisted of an initial 'pulse' followed by smaller steering corrections [S3B Fig; [9, 47, 48]]. Therefore, in our specific scenario the amount the driver turned the wheel in the initial steering response ($SWA_{Max}$) is a robust indicator of steering 'aggression' (or demand), and correlated highly with other measurements that have been used in the literature to characterise 'aggression' of steering response (e.g. Pearson's R: maximum steering wheel angle derivative = .87; steering wheel variability = .81). $SWA_{Max}$ was calculated by taking the difference between the steering wheel angle at disengagement and the maximum steering wheel angle in the 2 s window after takeover (S1 Fig). Trials, where drivers took over control with less than .25 s of the trial remaining, were excluded as after extensive inspection of individual steering traces it was judged that .25 s was too early for drivers to finish the initial steering correction (this removed 19 trials [1.2%], the mean time until $SWA_{Max}$ was .64 s, SD = .3 s).

The criticality at takeover ($TLC_T$) can be treated as a proxy for steering demand (i.e. how much steering is required). To examine whether cognitive load directly affects steering behaviour (rather than indirectly via slowed reaction times), $SWA_{Max}$ was modelled using both $TLC_T$ (*T* in Eq 8; coefficient $\gamma_T$) and ACMT (*L*; $\gamma_L$), including an interaction term ($\gamma_{TL}$). $SWA_{Max}$ is approximately lognormally distributed [cf. [49]], and appears related to $TLC_T$ via a power law (at low $TLC_T$ values $SWA_{Max}$ grows exponentially; Fig 4A). Taking the logarithm of both $SWA_{Max}$ and $TLC_T$ results in a strong linear relationship (Fig 4B). It is worth noting that there are nuances to interpreting the coefficients when the model is fitted in these log-log coordinates. On the arithmetic scale, the coefficients are multiplicative (see Eq 6) so they should be interpreted in terms of percentage change (see Materials and methods for more details).

The parameter means and 95% HDIs are given in Table 2, as well as the the estimated variability of the parameters between participants. The negative estimate of $\gamma_T$ has the effect that as $TLC_T$ tends towards zero, participants make larger steering adjustments ($SWA_{Max}$ tends towards infinity), and at large $TLC_T$ values, participants steer much less ($SWA_{Max}$ asymptotes at zero; see also Fig 4A & 4C). There is also a high likelihood that the presence of ACMT alters steering response. The parameter $\gamma_L$ is negative, causing a downward shift of intercept in log-log coordinates (Fig 4D). This can be interpreted in terms of percentage change on the arithmetic scale, such that when the ACMT is present steering response is *reduced* by around 12% (cf. $e^{\gamma_L}$ in Table 2). Though there is some uncertainty to the exact magnitude of this dampening effect (the 95% HDI range varies from 20% to 3% reduction), we can state with confidence that steering was attenuated when participants were engaged in the ACMT. The interaction term, $\gamma_{TL}$ is estimated to be close to zero, suggesting the ACMT acts primarily to shift the intercept rather than the slope of the regression line (Fig 4D).

## Discussion

This experiment was designed to investigate humans detecting and responding to silent failures of automated driving that occurred whilst steering around bending roads. The criticality of silent failures was manipulated to vary the required timing and magnitude of steering responses by the supervising driver to avoid leaving the road. The results showed that for less critical failures of automation, the drivers responded more slowly to the failure, but still with a

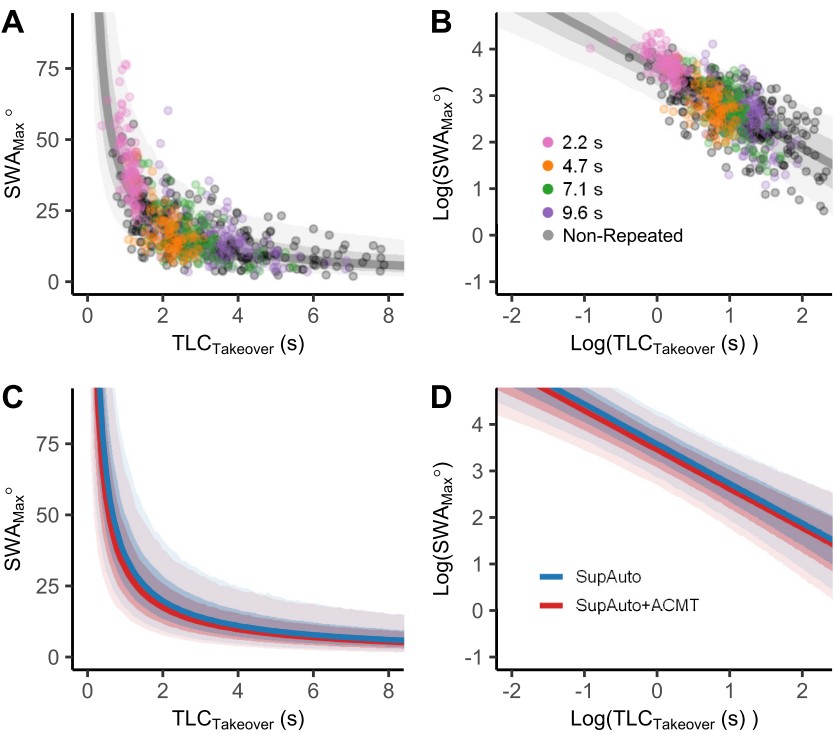

**Fig 4. SWA$_{Max}$.** In A) & B) SWA$_{Max}$ is plotted against TLC$_T$ for the SupAuto condition, shown in A) raw coordinates and B) log-log coordinates. The Repeated conditions are coloured in both A) & B) to indicate how the log-log coordinates transform the data. These data are overlaid on the SupAuto model mean and predictive intervals. The model regression lines and predictive bounds for 68.3% and 95.5% quantiles for SupAuto and SupAuto+ACMT are shown in panels C) & D).

higher safety margin (i.e. adopted a higher TLC at takeover), and were more variable in their timing of responses. Cognitive load was manipulated by adding an auditory task to some trials. When this additional load was present, drivers showed a small but consistent decrease in their adopted safety margin (i.e. adopted lower TLC values at takeover), and also displayed an increase in the variability of the timing of their responses. Whilst the magnitude of steering responses were scaled to the criticality at takeover, the added cognitive load acted to reduce the magnitude of steering responses.

The criticality of the failure conditions was varied to determine whether there was any concomitant adjustment in the timing of driver responses. If participants responded at a

**Table 2. Posterior means and 95% highest density intervals for parameters predicting the mean of ln(SWA$_{Max}$), $\mu$.**

| Fixed Effects | | | | | Random Effects | | | |
|---|---|---|---|---|---|---|---|---|
| Parameter | Description | Mean | Lower | Upper | $\sigma_{Parameter}$ | Mean | Lower | Upper |
| $\gamma_0$ [$e^{\gamma_0}$] | $\mu$ intercept [scaling constant on $e^{\mu}$] | 3.57 [35.67] | 3.49 [32.67] | 3.67 [39.27] | $\sigma_{\gamma_0}$ | .18 | .10 | .25 |
| $\gamma_T$ [$e^{\gamma_T}$] | ln(TLC$_T$) effect on $\mu$ [non-linearity of TLC$_T$ on $e^{\mu}$] | -.85 [.43] | -.94 [.39] | -.76 [.47] | $\sigma_{\gamma_T}$ | .17 | .11 | .26 |
| $\gamma_L$ [$e^{\gamma_L}$] | ACMT additive effect on $\mu$ [scaling effect of ACMT] | -.13 [.88] | -.22 [.80] | -.03 [.97] | $\sigma_{\gamma_L}$ | .14 | .06 | .24 |
| $\gamma_{TL}$ [$e^{\gamma_{TL}}$] | ACMT × TLC$_T$ effect on $\mu$ [scales the non-linearity when ACMT is present] | .01 [1.01] | -.06 [.94] | .10 [1.10] | $\sigma_{\gamma_{FL}}$ | .10 | .00 | .18 |

The exponentiated parameter for predicting $e^{\mu}$, the geometric mean on the arithmetic scale, are given in square brackets.

single TLC, then there would have been no change in $TLC_T$ across failure conditions (the regression line in Fig 2A would have been flat), whereas, if participants responded with consistent reaction time, a slope of 1 would have been expected (dashed lines in Fig 2A). The actual pattern of responses sat somewhere in-between. The safer response timings for less critical takeovers is consistent with studies examining planned failures [19, 22, 31, 50]. Furthermore, some automation studies on straight or low curvature highways have observed slower reaction times for less critical failures [4, 5, 31, 50]. The present findings demonstrate that this pattern holds for silent failures on bending roads, across a wide range of failure criticalities. The non-unity increase could have implications for the perceptual mechanisms underpinning how drivers decide when to intervene in silent failures [7]. The perceptual error at response (quantified by lower $TLC_T$ values) decreased with more gradual failures. Such behaviour could be explained by accounts of drivers responding to the *accumulated* perceptual error, equating integration of a small error over a long time with the integration of a large error over a short time, resulting in responses at smaller absolute error in less urgent situations (cf. [7, 9, 51]).

Though $TLC_T$ increased for less critical failures, $TLC_T$ values *decreased* due to slower responses when drivers were engaged with the auditory cognitive task. This result extends findings from previous drift-correction silent failure paradigms that found slower responses when using visual (watching movie clips compared to manual [12] and visual-motor [4] non-driving-related tasks. The results also agree with previous work on planned takeovers that shows reduced TLC [20] or TTC [19, 21, 52, 53] but also see [22]), and generally slower responses [5, 17–21], across a variety of secondary tasks. Slower responses when performing the ACMT does contradict Zhang et al. [5] who reported a negligible effect of primarily auditory tasks, but that meta-analysis aggregated across many planned takeover paradigms where a variety of secondary tasks are used, and drivers could intervene both longitudinally (by braking) and laterally (by steering). In contrast, the current study uses highly controlled conditions and many repetitions to precisely examine the effect of auditory cognitive load on steering behaviour across a wide range of silent failures.

The measures of central tendency we have discussed so far demonstrate broad shifts in the timing behaviour across conditions but do not indicate how variable responses were or whether variability changed. The results show that the variability of $TLC_T$ increased with $TLC_F$. An increase in variability for slower/less severe scenarios has been reported previously [4, 5, 40, 54], however, in the current study, the variability of response timing has been explicitly modelled using a hierarchical model. This approach allows the estimation of the relative contribution of within- and between-person variation. The biggest contributor to the spread of predicted $TLC_T$s is within-participant variability (61%), rather than between-participant variability (35%), meaning that trial-by-trial variation *within* individuals were greater than the difference in participant averages *between* individuals. The ACMT increased the spread of the TLC by ≈10%, but this increase is small compared with the estimated within- and between-subject variability. It should be noted that the sample size was relatively small, which can mean that the variance of random effects may be underestimated [55], or unduly influenced by the choice of prior [56]. Importantly, the width of the prior did not substantially alter the relative contributions to variability. Nevertheless, the *absolute* magnitude of the coefficients of variation should be taken only as an *approximate* indicator of scale for providing a useful benchmark for any mechanistic model attempting to incorporate stochasticity into predictions. Future work is needed on bigger samples and using heterogeneous scenarios to assess whether the estimated variability generalises.

Whilst the timing of driver responses detecting silent AV failures is important, a key aspect of the current manuscript is the examination of the magnitude of steering response (quantified

by $SWA_{Max}$). The results demonstrate that the relationship between $SWA_{Max}$ and $TLC_T$ can be captured using a power law: severe failures $SWA_{Max}$ tended towards large values, and less critical failures $SWA_{Max}$ tended towards zero. Some aspects of this finding have been previously discussed in the literature. Steering adjustments have been shown to be log-normally distributed, providing a rationale for modelling steering as a multiplicative control process [57]. Furthermore, some models of steering have related steering adjustments, specifically to TLC [37, 58]. However, to the authors' knowledge, the current study represents the first to empirically capture, with rigorous experimental control, the nature of the scaling relationship between $SWA_{Max}$ and TLC.

While the current study focuses on lateral control, previous research has linked TLC to longitudinal control, relating TLC to speed choice both empirically ([59], but see [60]) and in driver models [61]. Furthermore, models of braking behaviour have modelled brake strength as a linear function of the inverse of TTC [38, 62, 63], which is similar to the relationship found in the current study (the exponent of $TLC_T$ is estimated to be around -.85; a linear relationship to the inverse $TLC_T$ is equivalent to an exponent of -1). Though the precise magnitude of the estimated coefficients may be specific to this study (and the driver model used in simulator etc.), it seems that relating driver behaviour to indicators of remaining safety margins (e.g. TTC or TLC) is a promising avenue for developing driver models for silent failures.

The effect of the cognitive task on the timing of response has already been described above, however, the results also demonstrated that the magnitude of steering response was reduced when a cognitive load was added. A visually distracting task has been shown to increase $SWA_{Max}$ [12] in silent failures. However, they did not control for the possibility that slower reaction times caused conditions that then necessitated greater steering wheel corrections (see S2 Fig for the extent to which this applies to our scenario). To avoid this issue with the present dataset, instead of comparing condition averages of $SWA_{Max}$, $SWA_{Max}$ is predicted by $TLC_T$, therefore accounting for variation in the scenario at takeover. This method confirms that irrespective of the criticality at the time of response there was a general dampening of $SWA_{Max}$ due to added cognitive load. This finding would seem to contradict reports of improved lane keeping with added cognitive load (e.g. [24, 26]) that have been previously explained by cognitive load inducing a fallback to over-learned driving functions [23]. Instead of *enhancing* steering corrections, our results agree with reports of subdued steering action when a driver is cognitively loaded during planned takeovers ([29], note that this study used the same ACMT task as the current study). However, this apparent discrepancy could be reconciled if one considers that the task of detecting and responding to silent failures (and responding to cued handovers; [29]) will be a novel experience for most of, if not all, the participants. Therefore, non-loaded participants may have deployed cognitive control [30] to achieve good performance both at detecting failures and quickly reducing steering error. Cognitive load may have impaired these non-automatised aspects of the task [23], consequently reducing the effort made to steer quickly away from the road edges, which manifests in a dampened steering response. The same argument might also explain the delayed timing of response when loaded. An important outstanding question is how these effects translate to silent failures in real-world automated vehicles. If the effects of cognitive load are dependent on how well-learned the task is [23] then we might expect these effects to depend on the level of experience with automated vehicles (diminishing with increased experience). However, it takes many repetitions for a task to become automated [64], and AV failures are expected to be infrequent [65], reducing the opportunity for practice, therefore effects of cognitive load may persist despite growing AV use.

## Applied relevance

The patterns of behaviour described so far have considered the reliability of effects from an experimental perspective. One potential challenge could be that while scientifically interesting, the observed effects may be relatively minor with little real-world significance. One strength of using hierarchical Bayesian analysis methods is that they can be used to estimate the probability of particular consequences (namely the vehicle actually leaving the road) by sampling from the posterior predictive distribution implied by the estimated within- and between-subject variance (whilst accounting for uncertainty in the fitted parameters). This approach can be used to simulate regression coefficients for a range of unobserved "hypothetical" drivers. For each $TLC_F$ the simulated driver has a predicted mean and standard deviation of response, and from which practical safety implications can be derived.

An unambiguous marker for an unsafe takeover is how often the driver is predicted to exit the lane: $P(Exit)$. Trials with a negative $TLC_T$ indicate that the AV has left the road *before* the simulated driver takes control. However, this approach does not take into account turning arc so may miss responses that take over before leaving the road but still poses a real safety risk (in the current study were 9 instances where drivers exited the road *after* takeover). Therefore, it is sensible to include a 'point of no return' whereby $TLC_T$ is considered too small for the driver to stay within lane boundaries. It is difficult to be certain what the safety threshold should be, as it is likely to vary across individuals and the scenario. For example, in the current dataset the lowest $TLC_T$ observed for drivers that stayed *within* the lane was .46 s, yet there were five occasions where drivers exited the road despite $TLC_T >$ .46 *s* (mean $TLC_T$ for lane exits = .56 s, range = .25 s—.9 s). To avoid adopting a threshold that is too low, and therefore underestimate $P(Exit)$, we use a value of .5 s as the safety threshold in the applied simulations, but note that the choice of the threshold will affect $P(Exit)$ (S4 Fig).

Each simulated driver has an associated probability of exiting the road ($P(Exit)$; the proportion of trials with $TLC_T <$ .5 s). Therefore, from the posterior predictive distribution, the average $P(Exit)$ for the population can be estimated. Fig 5A shows the predicted $P(Exit)$ across different failure states. To provide a useful frame of reference for the applied relevance of these predictions, vertical lines are included in Fig 5A that represents the $TLC_F$ if an AV was to stop turning and travel straight ahead while on a bend (i.e. an off-tangent failure). The examples are classed as "rural roads" and "motorways" that adhere to the UK design standards for different UK highways [66, 67].

The model shows that $P(Exit)$ rises sharply as $TLC_F$ approaches zero (Fig 5A), though failures of this severity may be infrequent in the real-world since the road would need to be unusually narrow or tight, or the driver travelling well above the speed limit. Failure rates in the $TLC_F$ region 1.5–3 s (note the examples given in Fig 5A) could occur if, for example, the vehicle ceased turning and instead drifted along its longitudinal axis; failure rates where $TLC_F > 4$ *s* are likely to be very low curvature bends, or when the AV drifts very slowly (e.g. following the wrong line markings). Drivers are predicted to be safer when there is no additional cognitive load: e.g. for gradual failures ($TLC_F >$ 4 s), only around .5% of failures exit the road ($+2\sigma \approx$ 2%) whereas this estimate is around 1.5% ($+2\sigma \approx$ 4%) with added cognitive load. For more critical failures, $P(Exit)$ rises quickly, e.g. at $TLC_F = 2$ s, which could correspond to an off-tangent failure on a bend, $P(Exit)$ for SupAuto is 1.3% ($+2\sigma =$ 3.8%); for SupAuto+ACMT $P(Exit)$ is 4.4% ($+2\sigma =$ 10.2%). A potentially unintuitive aspect of Fig 5 is that $P(Exit)$ does not continue to fall as failures become more gradual. This behaviour emerges due to modelling the within-individual variability with both $TLC_F$ and ACMT acting as linear predictors. Whilst this choice provides a good fit of the data, it seems implausible that variability would continue

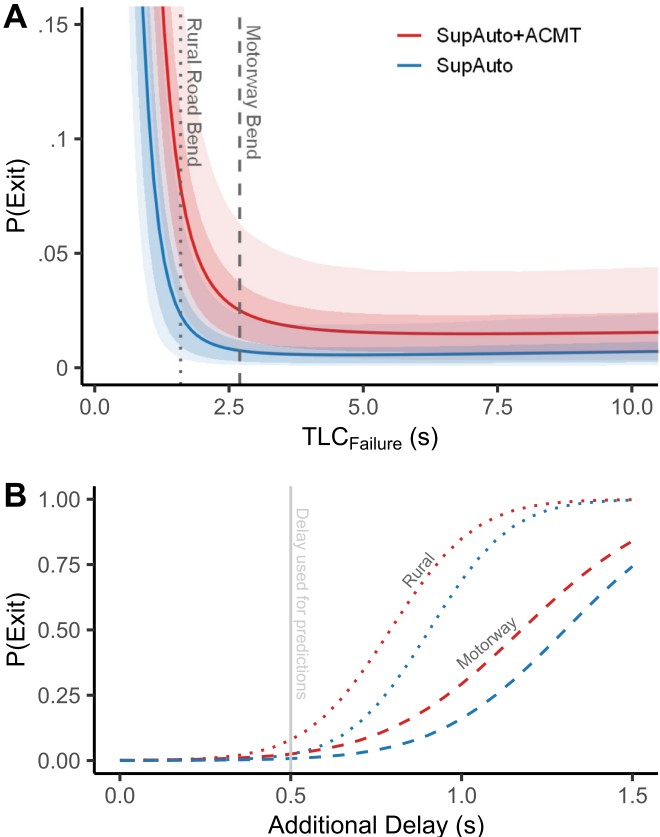

**Fig 5.** A) Predicted probability of exiting the road before disengaging the automation when loaded (SupAuto + ACMT) and not loaded (SupAuto). Specifically, $P(Exit)$ refers to the proportion of simulated failures with a $TLC_T$ of $<.5$ s. Solid lines represent the average prediction, and bounds are the 68.3 and 95.5% quantiles. Dashed lines represent the $TLC_F$ for off-tangent failures on typical bends on a single lane carriageway (Radius = 500 m, speed = 60 mph (26.82 ms), lane width = 3.65 m, $TLC_F$ = 1.6 s) and a multiple-lane motorway (Radius = 2000 m, speed = 70 mph (26.82 ms), lane width = 3.65 m, $TLC_F$ = 2.7 s). B) How $P(Exit)$ increases when further delays are included in the predictions. The mean estimates are plotted for the examples shown in panel A: bends on a motorway (dashed line) and rural road (dotted line). The vertical grey line shows the delay value used for the predictions presented in panel A.

to rise in this way. More likely, there is an upper bound on $\sigma$, but due to the censored nature of the data (limited trial length), it was not possible to effectively model this upper bound.

The predictions in Fig 5 help to illustrate the potential benefits of using generative models for regression analysis in this domain. There are several reasons why drivers may have detected failures more quickly in the present highly-controlled experiment, compared to noisy real-world driving conditions: there was no traffic [35], participants experienced many failure repetitions [20, 22, 33, 68], and gaze was directed forwards because there were few visual distractions [34]. Relaxing any of these constraints could increase the predicted $P(Exit)$ (Fig 5B & S4 Fig). It should be noted that it is also possible that detection of AV failure could have been artificially *slowed* by the lack of vestibular cues (we used a fixed-based simulator) and no vehicular sounds (which prevented interference with the ACMT task), both of which can contribute to successful driving [69] and could provide a signal that there has been AV failure.

A further limitation of applying the model relates to taking $TLC_T$ as a direct indicator of whether the driver is safely in control of the vehicle. Specifically, $TLC_T$ only considers the timing of when the driver takes over control. While we account for changes in the trajectory *after* disengagement by applying a delay to $TLC_T$, the method would be improved by explicitly

including a model of how drivers steer during takeovers, and also by incorporating vehicle dynamics into the TLC calculation [e.g. vehicle extent and wheel slip; [36]). As yet, adequate models of this do not exist [3]. It is hoped that the present detailed examination of how drivers detect and respond to silent failures will usefully inform the development of such models.

Most of the limitations described are likely to increase $P(Exit)$, so the authors caution that the predictions presented in Fig 5 should be considered as the best-case scenario, and treated as a lower-bound estimate for the real-world safety risk of silent failures. Further research is still needed to examine factors that might delay or impair the driver's corrective manoeuvre to silent failures. To highlight the importance of these efforts, Fig 5B hypothesises how—based on the current dataset—additional delay might increase $P(Exit)$. The relationship is non-linear, with increasing delay corresponding to a rapidly increasing $P(Exit)$, and more pronounced for more critical failures (i.e. the 'Rural Road' compared to the 'Motorway'). Fig 5B shows that even a relatively small increased delay for Fig 5 increases $P(Exit)$ to worrying levels (see also S4 Fig). As an example, consider for a moment trying to account for the predictable nature of the current experiment. Drivers who were faced with unpredictable planned takeovers have been estimated to be around 1 s slower than drivers who had previously experienced (and therefore will have some expectation of) a planned takeover [5]. A further 1 s delay (giving a safety threshold of 1.5 s) would mean more than 75% of AV failures result in lane exits for the specified scenarios (Fig 5B).

## Conclusion

This manuscript examines silent failure detection and steering responses to 28 failure conditions. Driver behaviour is highly dependant on failure criticality. Drivers take over control with longer response times and higher safety margins for less severe failures, yet they are also more variable. The magnitude of the steering response is scaled to the criticality. An auditory secondary task caused drivers to take over later, make more variable responses, and also make smaller initial steering corrections.

Using bayesian hierarchical models, criticality (TLC) at takeover was ably predicted using a gaussian distribution where the mean and standard deviation both increased as failure severity decreased. Furthermore, the magnitude of steering response was related to the criticality at takeover through a power law, with highly critical takeover producing increasingly large corrections and less critical takeovers tending towards minimal corrections. Hierarchical modelling of both the mean and variability of TLC showed that both within- and between-individual variability should be taken into account when predicting safety boundaries, and also when developing mechanistic models for virtual testing. These methods allow for applied simulations of hypothetical failures, providing a lower-bound estimate of the probability that a driver would exit the road before taking over control of an automated vehicle that has failed. The lower-bound is not negligible (about 1/100 failures, rising quickly for critical failures), and the probability is expected to rise rapidly when additional sources of delays are incorporated (e.g. due to traffic, or surprising failures not tested in this manuscript). This modelling should be a cause for concern when considering the widespread plans to adopt AV systems.

## Materials and methods

### Open science

The raw data, analysis scripts, and experiment code are freely available on the Open Science Framework [70], as well as a pre-registration [71]. These data were collected according to the pre-registration. The preregistration describes the planned analyses both of steering and gaze data, however, due to the scale of analysis required to thoroughly investigate each set of

behaviours, we have chosen to report here the findings related to steering responses and create a separate manuscript to report gaze behaviours.

### Participants

Twenty staff and students (7 Females) of the University of Leeds volunteered to participate in the present study (Mean age = 25.2 years, range = 20-32 years). Participants had normal or corrected to normal, hearing and sight. Most (N = 17) participants had UK driving licences, for an average of 6 years. Participants were paid £10 for their time (1 hour). The study was approved by the University of Leeds Research Ethics Committee (Ref: PSC-564) and complied with the guidelines set out in the declaration of Helsinki. Written informed consent was given.

### Driving simulator

The experiment took place in a fixed-based driving simulator, with stimuli back-projected onto a large projection screen (field of view 89˚ x 58˚) with black surroundings. Participants sat on a height-adjustable seat with eye position 1.2 m high and 1 m from the display. The experiment was run on a desktop PC with Intel i7 3770 (3.40 GHz). Display refresh and data recording rates were synchronized at 60 Hz. The stimuli were generated using Vizard 5 (WorldViz, Santa Barbara, CA), a Python-based software for rendering perspective correct virtual environments. Participants steered using a force-feedback wheel (Logitech G27, Logitech, Fremont, CA). The road geometry across all conditions began with a straight section of 16 m length (2 s), followed by a constant curvature bend of 80 m radius (either leftwards or rightwards). The road width was 3 m. The road was rendered using a semi-transparent grey texture. The ground plane of the virtual environment was textured with 'Brownian noise' (as per [72], Fig 1E), which has been shown to elicit similar gaze behaviours to on-road driving [72]. Vehicle speed was kept constant at 8 ms$^{-1}$ ($\approx$ 18 mph).

### Silent failure selection

Repeated trials had the same automated driving trajectory and the failure was introduced into the simulation at the same time (6 s into the trial; the *onset* time). The visual stimulus produced was therefore identical in each repetition. The most rapid (TLC$_F$ = 2.23 s) was a 'tangential' silent failure (the vehicle continued along its longitudinal axis), whereas the most gradual silent failure (TLC$_F$ = 9.55 s) would not cause the vehicle to leave the road within the period of the trial. The middle failures severities (TLC$_F$ = 4.68 s, 7.12 s) were equally spaced between these two most extreme failures so that the parameter space of TLC$_F$ was explored. The yaw-rate offsets for the Repeated failures were 5.73, 1.20, .52, and.30˚/s). To avoid easily detectable step shifts in yaw-rate the bias was introduced via a smooth step function (over.5 s) that ensured that the derivative of yaw-rate was smooth.

We complemented repeated trials with non-repeated trials selected from a wider range of TLC$_F$s (from a range of 2.95 s to 19.51 s). Within the non-repeated trials, we also varied the automated driving trajectory (from a pool of four pre-recorded trials), the failure onset time (from a range of 5 s to 9 s), and whether the direction of failure was oversteering or understeering (set to understeer 70% of the time). The non-repeated trials needed to be unpredictable and also to adequately explore the space. Therefore, the parameters were chosen using a 4-dimensional Sobol sequence—a convenient way of generating a quasi-random string of values that adequately explores a range of values. In total, there were 28 failure conditions.

## Cognitive load: Auditory distraction task

During each trial the auditory equivalents of the visual targets were presented amongst a stream of auditory distractor items, that occurred at a random interval varying between 1.0 s— 1.5 s (in 0.1 s steps; Fig 1D & 1E). The task was designed so that drivers could respond to the ACMT (using their thumbs) and take over control of the vehicle (using their fingers) without moving their hands, and use whichever hand they wished for either task, so the ACMT should have a minimal effect (if any) on takeover timings. The ACMT task continued until the end of each trial (i.e. through both automated and manual periods). At the end of each trial participants also reported how many of each target letter they thought they had detected. Reporting was electronically recorded using the steering wheel, then participants confirmed their selection by clicking the paddle shifters situated behind the wheel.

All participants did well on the ACMT task (responding appropriate 92.6% of the time, with a mean reaction time of .75 s), suggesting high engagement. We found little evidence of trade-offs: while participants, in general, were marginally slower (and less correct) at responding to the ACMT in SupAuto+ACMT compared to baseline ACMT performance, we did not find that drivers that performed worse on the ACMT responded substantially more quickly to automation failures.

## Procedure

Participants experienced three 50 s long practice laps on a sinusoidal track with bend radii of 60 m. On the first lap, drivers had manual control. The second and third practice laps began in automation and the participant was instructed to supervise and take over control by pressing the gear pads when they were ready to do so. This ensured that participants were familiar with the simulator dynamics, the automation driving, and the takeover method. Participants also practiced the ACMT (without driving) until they were comfortable with the instructions.

The SupAuto (supervising automation) task consisted of a series of discrete trials (half bending leftwards and half rightwards) where an automated vehicle trajectory was simulated by replaying a pre-recorded trajectory of a well-practiced driver that steered smoothly and kept close to the midline (Fig 1). During automation, participants kept their hands loosely on the wheel, which moved in correspondence to the visual scene. The takeover was initiated by pressing a paddle shifter and was confirmed with a high-pitched (480Hz, 200ms) tone. Control transfer was immediate. Each trial began with a 2 s pause without vehicle motion, during which time the wheel was automatically re-centred. The locomotor component of each trial was 15 s, after which the scene was reset (in SupAuto) or the ACMT task was shown (Fig 1E). The time taken for participants to submit their estimated counts of targets at the end of each ACMT trial was unrestricted.

Baseline ACMT measures (without driving) were taken before and after the driving blocks of trials so that participant trade-offs (between the ACMT and failure detection) could be assessed. Participants conducted the experiment in four blocks: ACMT only, SupAuto, SupAuto+ACMT, ACMT only. The SupAuto and SupAuto+ACMT blocks were counterbalanced across participants. Within each block conditions were randomly interleaved. Each participant completed 192 trials (96 each for SupAuto and SupAuto+ACMT).

## Model fitting

Repeated and Non-Repeated trials were pooled into the one model fitting. Both models were fitted using Hamiltonian Monte Carlo in Stan, using the R package brms [73]. Weakly informative priors were chosen, but the results for both $TLC_T$ and $SWA_{Max}$ were robust to changes in prior specifications. The final models were arrived at through iterative increases in

complexity, with model comparisons being made with leave-one-out cross validation [which aims to counter over-fitting by estimating out-of-sample prediction error, [74]). Additional terms were only kept if they decreased prediction error and had a clear interpretation.

## Modelling TLC at takeover

$TLC_T$ cannot be higher than $TLC_F$ (the 1:1 line in Fig 2A), or lower than $TLC_{End}$ (the gold bars in Fig 2A). Therefore, $TLC_T$ is modelled as a normal distribution, truncated (capped) by $TLC_F$ at one end and censored (i.e. the measurement is limited but the measured distribution can in theory extend past the censored value) by $TLC_{End}$ at the other. The between-participant covariation of predictors is modelled with a multivariate gaussian specified by covariance matrices $\mathbf{S}_\beta$, $\mathbf{S}_\alpha$. The distributional model for $TLC_T$ is given below:

$$TLC_{T_i} \quad \sim Normal(\mu_i, \sigma_i) \tag{1}$$

$$\mu_i \quad = \beta_{0_j} + \beta_{F_j} F_i + \beta_{L_j} L_i + \beta_{FL_j} F_i L_i \tag{2}$$

$$\ln(\sigma_i) \quad = \alpha_{0_j} + \alpha_{F_j} \ln(F_i) + \alpha_{L_j} L_i \tag{3}$$

$$\begin{bmatrix} \beta_{0_j} \\ \beta_{F_j} \\ \beta_{L_j} \\ \beta_{FL_j} \end{bmatrix} \sim MultivariateNormal \left( \begin{bmatrix} \beta_0 \\ \beta_F \\ \beta_L \\ \beta_{FL} \end{bmatrix}, \mathbf{S}_\beta \right) \tag{4}$$

$$\begin{bmatrix} \alpha_{0_j} \\ \alpha_{F_j} \\ \alpha_{L_j} \end{bmatrix} \sim MultivariateNormal \left( \begin{bmatrix} \alpha_0 \\ \alpha_F \\ \alpha_L \end{bmatrix}, \mathbf{S}_\alpha \right) \tag{5}$$

Where $i$ indicates the condition and $j$ indicates the participant. $\mathbf{S}_\beta$ & $\mathbf{S}_\alpha$ are covariance matrices centred on the population coefficient values. Note that the logarithmic link function on $\sigma_i$ means that the linear predictors are multiplicative:

$$\sigma_i = e^{\alpha_0} F_i^{\alpha_F} e^{\alpha_L} \tag{6}$$

In exponentiated form the formula takes on a pleasing interpretation [45]. $e^{\alpha_0}$ is a constant that scales $F_i^{\alpha_F}$. The exponent $\alpha_F$ allows flexible modelling of non-linear trends (the linear case is $\alpha_F = 1$). When the ACMT task is present $e^{\alpha_L}$ acts as another constant that increases or decreases by a percentage. The *scaling* of variability (rather than dealing in *absolute* terms) due to cognitive load is intuitive and generalisable.

## Modelling maximum steering wheel angle

The distributional model for $SWA_{Max}$ is given below:

$$\ln{(SWA_{Max})_i} \quad \sim Normal(\mu_i, \sigma) \tag{7}$$

$$\mu_i \quad = \gamma_{0_j} + \gamma_{T_j}\ln{(T_i)} + \gamma_{L_j}L_i + \gamma_{TL_j}\ln{(T_i)}L_i \tag{8}$$

$$\begin{bmatrix} \gamma_{0_j} \\ \gamma_{T_j} \\ \gamma_{L_j} \\ \gamma_{TL_j} \end{bmatrix} \sim MultivariateNormal\left( \begin{bmatrix} \gamma_0 \\ \gamma_T \\ \gamma_L \\ \gamma_{TL} \end{bmatrix}, \mathbf{S} \right) \tag{9}$$

Where $i$ indicates the condition, $j$ the participant, and $\mathbf{S}$ the covariance matrix that allows coefficients to covary across participants.

As noted in the main text, on the arithmetic scale (i.e. exponentiated form) the coefficients are multiplicative. Furthermore, in contrast to when a logarithmic link is used (Eq 3), a complete log-transform of $SWA_{Max}$ means that when the model's predictions are de-transformed (exponentiated) to be on the arithmetic scale (i.e. the original units) the distribution of errors is multiplicative rather than additive [75]. Furthermore, the exponent of $\mu_i$ (which is an estimator for $\frac{1}{N}\sum\ln{(SWA_{Max})}$) corresponds to the *geometric* mean (which in this case is also the median value) on the arithmetic scale [76].

These characteristics are potentially useful: steering control has previously been modelled using multiplicative control inputs [57], and variability in the motor system is considered to be scaled to the size of the control signal [77, 78], thus both sensory and motor noise have been modelled as multiplicative when controlling a vehicle (e.g. [9, 79]).

## Supporting information

**S1 Fig. Sample steering wheel trace and identification of $SWA_{Max}$.** $SWA_{Max}$ is the difference between the initial steering wheel angle and the maximum steering wheel angle, taken within a 2 s time window.
(EPS)

**S2 Fig. Examining the relationship between $SWA_{Max}$ and reaction time.** Plotted are the four Repeated $TLC_F$ conditions. In every failure condition, there is a strong positive correlation between RT and $SWA_{Max}$. Pearson's R values range from.53 to.68 (mean = .61). The marginal means and standard deviations for RT and $SWA_{Max}$ are shown as dots close to their respective axis. The ACMT (SupAuto+ACMT) consistently slows reaction times. The average difference between SupAuto+ACMT and SupAuto conditions (averaging across each participant's mean difference between median RTs) is.19 s (SD = .37; one sample t-test comparing to zero difference: $t(18) = -2.26$, $p = .04$). Given the strong correlations one might expect this slowing to translate to $SWA_{Max}$ but in fact the condition averages for SupAuto and SupAuto+ACMT are approximately equal. The average difference between $SWA_{Max}$ for SupAuto+ACMT and SupAuto conditions is only.21˚ (SD = 4.0; one sample t-test comparing to zero difference: $t(18) = .22$, $p = .83$).
(EPS)

**S3 Fig. Examining the relationship between lane position and reaction time.** Plotted are the four Repeated $TLC_F$ conditions. In every failure condition there is a very strong positive correlation between RT and Lane Position. Pearson's R values range from.72 to.98 (mean = .87), and are generally closer to one for more gradual failures. The marginal means (dots) and standard deviations (lines) for RT and Lane Position are shown close to their respective axis. The ACMT (SupAuto+ACMT) consistently slows reaction times. The average difference between SupAuto+ACMT and SupAuto conditions (averaging across each participant's mean difference between median RTs) is.19 s (SD = .37; one sample t-test comparing to zero difference: $t(18) = -2.26$, $p = .04$). This appears to propagate into differences in Lane Position, since on average drivers edged.1 m (SD = .1) closer to the road edge in SupAuto+ACMT (one sample t-test comparing to zero difference: $t(18) = -4.28$, $p < .001$).
(EPS)

**S4 Fig. Predicted probability of exiting the road before disengaging the vehicle when loaded (SupAuto+ACMT) and not loaded (SupAuto) with additional delays, from 0–1 s (shown in panel labels).**
(EPS)

## Acknowledgments

Thanks to Oscar Giles for his contribution to the Python code used for steering wheel automation, and advice on the analytical approaches used in this paper.

## Author Contributions

**Conceptualization:** Callum Mole, Richard Romano, Natasha Merat, Gustav Markkula, Richard Wilkie.

**Formal analysis:** Callum Mole, Jami Pekkanen.

**Funding acquisition:** Callum Mole, Richard Romano, Natasha Merat, Gustav Markkula, Richard Wilkie.

**Methodology:** Callum Mole, Jami Pekkanen, William Sheppard, Gustav Markkula, Richard Wilkie.

**Project administration:** Richard Wilkie.

**Resources:** Richard Wilkie.

**Software:** Callum Mole, Jami Pekkanen.

**Supervision:** Gustav Markkula, Richard Wilkie.

**Visualization:** Callum Mole, Jami Pekkanen.

**Writing – original draft:** Callum Mole, Jami Pekkanen, Gustav Markkula, Richard Wilkie.

**Writing – review & editing:** Callum Mole, William Sheppard, Tyron Louw, Richard Romano, Natasha Merat, Gustav Markkula, Richard Wilkie.

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
