## [Decision Letter · Decision Letter 0]

3 Sep 2020

PONE-D-20-24803

Predicting takeover response to silent automated vehicle failures

PLOS ONE

Dear Dr. Mole,

Thank you for submitting your manuscript to PLOS ONE. After careful consideration, we feel that it has merit but does not fully meet PLOS ONE’s publication criteria as it currently stands. Therefore, we invite you to submit a revised version of the manuscript that addresses the points raised during the review process.

We look forward to receiving your revised manuscript.

Kind regards,

Feng Chen

Academic Editor

PLOS ONE

Journal Requirements:

2. We noted in your submission details that a portion of your manuscript may have been presented or published elsewhere.

"A version of the paper has been released as a preprint."

Please clarify whether this publication was peer-reviewed and formally published. If this work was previously peer-reviewed and published, in the cover letter please provide the reason that this work does not constitute dual publication and should be included in the current manuscript.

Reviewers' comments:

Reviewer's Responses to Questions

**Comments to the Author**

1. Is the manuscript technically sound, and do the data support the conclusions?

Reviewer #1: Yes

Reviewer #2: Yes

2. Has the statistical analysis been performed appropriately and rigorously? 

Reviewer #1: Yes

Reviewer #2: Yes

3. Have the authors made all data underlying the findings in their manuscript fully available?

Reviewer #1: Yes

Reviewer #2: Yes

4. Is the manuscript presented in an intelligible fashion and written in standard English?

Reviewer #1: Yes

Reviewer #2: Yes

5. Review Comments to the Author

Reviewer #1: 1. It is recommended that pictures of the driving simulator, the simulated driving route and the simulated driving scenario, as well as a flow chart of how the experiment was carried out, be added to enhance the reader's knowledge of the experiments carried out. The figure of the experimental scenario given in this paper shows that the experimental scenario differs greatly from the real road conditions, how to ensure that the results of the experiment are meaningful in this experimental state?

2. In this paper, reaction time is a very important parameter, but the definition of reaction time in this paper is vague, so it is recommended to clearly define the reaction time and explain its practical significance.

3. As the experiment progresses, the driver gradually adapts to the simulated driving scenario, producing a certain learning effect, when the driver may become more sensitive or more sluggish to the silent failure stimulus. It is proposed to explain how this paper is a scientifically based experimental approach to reduce the impact of driver learning effects on experimental results.

4. The title of this paper is “Predicting takeover response to silent automated vehicle failures”, therefore it is suggested that the description of the key performance of the predictive model be added to the conclusion section as appropriate to echo the theme of this paper and to enable the reader to quickly understand the key findings of this study in predicting response.

5. SI Figure 3 lacks a quantitative description of "The marginal means (dots) and standard deviations (lines) for RT and Lane Position", and given the statistical importance of the mean in describing the state, it is recommended that the magnitude of this statistical value be supplemented with an appropriate analysis of the value.

Reviewer #2: The topic of this paper is interesting and important. The methods sound. The results are meaningful and useful. There are several suggestions to improve this paper.

1. More information of the participants is needed, for example, the driving experience.

2. The structure of this paper is not so formal.

3. One table of the statistical information of the results is suggested.

4. One paper about the driving simulator experiment of the the steering performance under sudden situation maybe is useful for this paper.

[1] "Examining the safety of trucks under crosswind at bridge-tunnel section: A driving simulator study”, Tunnelling and Underground Space Technology, 2019, 92, 103034.

6. PLOS authors have the option to publish the peer review history of their article (what does this mean?). If published, this will include your full peer review and any attached files.

Reviewer #1: No

Reviewer #2: No

---

## [Author Response · Author response to Decision Letter 0]

21 Oct 2020

Reviewer #1: 1.

It is recommended that pictures of the driving simulator… and the simulated driving scenario,

We agree that it would be illustrative to include a picture of the driving simulator. Unfortunately, we do not have an up-to-date picture, and our laboratories currently remain closed to us as the University does not yet deem them covid-secure. If a picture is required we can investigate with the University management.

… the simulated driving route, and the simulated driving scenario…

The driving route and stimuli is described clearly in the methods: “The road geometry across all conditions began with a straight section of 16 m length (2 s), followed by a constant curvature bend of 80 m radius (either leftwards or rightwards). The road width was 3 m. The road was rendered using a semi-transparent grey texture. The ground plane of the virtual environment was textured with ‘Brownian noise’ (as per 72, Fig 1E), which has been shown to elicit similar gaze behaviours to on-road driving (72). Vehicle speed was kept constant at 8 ms­-1 (≈18 mph).”

Additionally, throughout the manuscript we take pains to describe the simulated driving scenario in considerable detail, including in multiple graphics (see Fig 1A & B; Fig1 Caption, Experiment, Silent Failures Selection). These descriptions were tested on three non-scientists to check for clarity and understanding. In all three tests the layperson was able to articulate, without prompts, the driving simulator scenario back to the first author.

In previous drafts of the manuscript we included a birds-eye view of some sample trajectories, which might be the type of graphic that the reviewer is suggesting. We initially thought that it would be instructive, but we found that the scale needed to include the full track made it difficult to see the shape of trajectories, so in the end we decided that these graphics were unhelpful because the track could be easily described with words (straight section followed by a constant curvature bend).

…as well as a flow chart of how the experiment was carried out…

In Fig 1 we described in detail the procedure of the experiment. The manuscript already has a large number of figures, and we are reluctant to add more. We realise that some details were placed in figure captions that may have been confusing if one was to read only the Procedure section. We have now added to the procedure to clarify that the trials were done as a sequence: “The locomotor component of each trial was 15 s, after which the scene was reset (in SupAuto) or the ACMT task was shown (Fig 1E).”

…be added to enhance the reader's knowledge of the experiments carried out. The figure of the experimental scenario given in this paper shows that the experimental scenario differs greatly from the real road conditions, how to ensure that the results of the experiment are meaningful in this experimental state?

We feel that a strength of our approach is the high degree of experimental control over the visual stimuli and reliable and repeatable conditions. This necessarily comes at the expense of ecological validity since real world driving is highly varied and variable. We deliberately constrained the visual stimuli so that the only sources of perceptual information were the road edges and the optic flow from the ground texture. By removing extraneous features that could serve as possible distractions and gaze fixation candidates, we are able to assess the perceptual-motor behaviour more rigorously, rather than including spurious gaze behaviours that would confound interpretation (e.g. during less critical failures drivers may look to irrelevant scene objects rather than the road ahead, therefore delay takeover due to not looking rather than due to accumulating perceptual error more slowly).

In the manuscript we acknowledge the limitations in the following section of the discussion:

“The predictions in Fig 5 help to illustrate the potential benefits of using generative models for regression analysis in this domain. There are several reasons why drivers may have detected failures more quickly in the present highly-controlled experiment compared to noisy real-world driving conditions: there was no traffic (35), participants experienced many failure repetitions (33; 68; 22; 20), and gaze was directed forwards because there were few visual distractions (34). Relaxing any of these constraints could increase the predicted P(Exit) (Fig 5B & SI Fig 4). It should be noted that it is also possible that detection of AV failure could have been artificially slowed by the lack of vestibular cues (we used a fixed-based simulator) and no vehicular sounds (which prevented interference with the ACMT task), both of which can contribute to successful driving (69) and could provide a signal that there has been AV failure”

2. In this paper, reaction time is a very important parameter, but the definition of reaction time in this paper is vague, so it is recommended to clearly define the reaction time and explain its practical significance.

We agree that reaction time is an important metric in the field. However, we also contend that the literature places too much emphasis on reaction time, and instead should report contextualising metric such as time-to-line-crossing (TLC; an argument which we make in the introduction). Our manuscript therefore uses TLC as the primary metric.

That being said, both TLC and RT are related in almost all real-world scenarios (though the mapping depends on the context). A strength of the current experimental design is that one can be derived directly from the other (I.e. TLC at takeover = TLC at failure – RT). We now include additional clarification at the beginning of the section ‘Detecting Failures: TLC at Takeover’ that “The timestamp of when the driver pulled the paddle shifter behind the steering wheel was taken as the takeover moment”. TLC at failure corresponds to the TLC at the time when the failure was introduced (the failure onset; this is described in the manuscript). Therefore, the reaction time is this timestamp minus the failure onset time.

3. As the experiment progresses, the driver gradually adapts to the simulated driving scenario, producing a certain learning effect, when the driver may become more sensitive or more sluggish to the silent failure stimulus. It is proposed to explain how this paper is a scientifically based experimental approach to reduce the impact of driver learning effects on experimental results.

The experimental blocks (SupAuto; SupAuto+ACMT) were counterbalanced, and the trials within each block were randomly interleaved (we have now added a clarifying sentence – “Within each block conditions were randomly interleaved.” in the Procedure). Therefore, though learning/fatigue effects within each participant might be expected, these would not have systematically mapped on to specific conditions, so is not a confound in the interpretation of our results. We specifically highlight the possibility of learning effects in the last couple of sentences in the discussion:

“As an example, consider for a moment trying to account for the predictable nature of the current experiment. Drivers who were faced with unpredictable planned takeovers have been estimated to be around 1 s slower than drivers who had previously experienced (and therefore will have some expectation of) a planned takeover (5). A further 1 s delay (giving a safety threshold of 1.5 s) would mean more than 75% of AV failures result in lane exits for the specified scenarios (Fig 5B).”

4. The title of this paper is “Predicting takeover response to silent automated vehicle failures”, therefore it is suggested that the description of the key performance of the predictive model be added to the conclusion section as appropriate to echo the theme of this paper and to enable the reader to quickly understand the key findings of this study in predicting response.

Thank you, we agree that in our attempts at succinctness we may have made the key conclusions hard to parse quickly. The second paragraph of the conclusion concerns the predictive model, and now reads as follows (changes highlighted):

“Using bayesian hierarchical models, criticality (TLC) at takeover was ably predicted using a gaussian distribution where the mean and standard deviation both increased as failure severity decreased. Furthermore, the magnitude of steering response was related to the criticality at takeover through a power law, with highly critical takeover producing increasingly large corrections and less critical takeovers tending towards minimal corrections. Hierarchical modelling of both the mean and variability of TLC showed that both within- and between-individual variability should be taken into account when predicting safety boundaries, and also when developing mechanistic models for virtual testing. These methods allow for applied simulations of hypothetical failures, providing a lower-bound estimate of the probability that a driver would exit the road before taking over control of an automated vehicle that has failed. The lower-bound is not negligible (about 1/100 failures, rising quickly for critical failures), and the probability is expected to rise rapidly when additional sources of delays are incorporated (e.g. due to traffic, or surprising failures not tested in this manuscript). This modelling should be a cause for concern when considering the widespread plans to adopt AV systems.”

5. SI Figure 3 lacks a quantitative description of "The marginal means (dots) and standard deviations (lines) for RT and Lane Position", and given the statistical importance of the mean in describing the state, it is recommended that the magnitude of this statistical value be supplemented with an appropriate analysis of the value.

To enable the reader to better assess the magnitude of the differences between conditions we now add one-sample t-tests comparing the differences between cognitive load conditions to zero, for RT, steering wheel angle, and lane position (i.e. for both SI Fig 2 and SI Fig 3). These are highlighted in the manuscript.

Reviewer #2: The topic of this paper is interesting and important. The methods sound. The results are meaningful and useful. There are several suggestions to improve this paper.

1. More information of the participants is needed, for example, the driving experience.

Thank you. We report that 17/19 participants had driving licenses, for an average of 6 years. Unfortunately the length of license is the only information on driving experience we have. That being said, we nevertheless do not feel that considerable driving experience is an important aspect of the study, or that our pattern of results could be explained by driver (in)experience. The participants only needed to control a steering wheel, and monitor when to take over of a vehicle. In our highly controlled scenario this behaviour is akin to a simple perceptual-motor error detection task, which is quickly learned. There are no traffic rules or complex driving situations to negotiate, for which experience might be beneficial. Furthermore, we offer practice with the driving simulator, and our highly controlled experimental conditions allow us to quantify (and control for) individual participant variability.

2. The structure of this paper is not so formal.

We agree that the structure of our paper uses the Results-First format, which is atypical to many papers that describe the Methods before the Results. We chose this format because a quick reader may obtain the core understanding by Fig1 and reading the Results sections. The Materials and Methods section provide more detail for the interested reader, but are non-essential for the core flow of the manuscript. Instead of breaking up the flow from the Introduction to the Results, we chose to put the Methods at the end.

3. One table of the statistical information of the results is suggested.

Thank you. We have tried to produce a single table with all the results, however, we found that the single large table was difficult to understand, since the model parameters have different interpretations depending on the measure. Further, since the first results section (TLC) is quite large, a single table will be quite spatially distant when the second results section (SWAMax) is reached, causing difficulty for the reader referring back. Since PlosOne allows two tables, we think that it is clearer to the reader to separate the metrics into two tables and hope you agree with our rationale.

4. One paper about the driving simulator experiment of the the steering performance under sudden situation maybe is useful for this paper.

[1] "Examining the safety of trucks under crosswind at bridge-tunnel section: A driving simulator study”, Tunnelling and Underground Space Technology, 2019, 92, 103034.

We thank the reviewer for highlighting this interesting paper, which the authors had not seen. However, though the paper concerns driving responses to sudden perturbations, the drivers are in manual control the entire drive. This scenario has fundamental differences to monitoring an automated vehicle, so due to the long reference list (we already have 79 references) we have decided to omit this paper from the manuscript in favour of similar papers that are more relevant to silent failures of automation.

---

## [Decision Letter · Decision Letter 1]

10 Nov 2020

Predicting takeover response to silent automated vehicle failures

PONE-D-20-24803R1

Dear Dr. Mole,

We’re pleased to inform you that your manuscript has been judged scientifically suitable for publication and will be formally accepted for publication once it meets all outstanding technical requirements.

Kind regards,

Feng Chen

Academic Editor

PLOS ONE

Additional Editor Comments (optional):

Reviewers' comments:

Reviewer's Responses to Questions

**Comments to the Author**

1. If the authors have adequately addressed your comments raised in a previous round of review and you feel that this manuscript is now acceptable for publication, you may indicate that here to bypass the “Comments to the Author” section, enter your conflict of interest statement in the “Confidential to Editor” section, and submit your "Accept" recommendation.

Reviewer #1: All comments have been addressed

Reviewer #2: All comments have been addressed

2. Is the manuscript technically sound, and do the data support the conclusions?

Reviewer #1: Yes

Reviewer #2: Yes

3. Has the statistical analysis been performed appropriately and rigorously? 

Reviewer #1: Yes

Reviewer #2: Yes

4. Have the authors made all data underlying the findings in their manuscript fully available?

Reviewer #1: Yes

Reviewer #2: Yes

5. Is the manuscript presented in an intelligible fashion and written in standard English?

Reviewer #1: Yes

Reviewer #2: Yes

6. Review Comments to the Author

Reviewer #1: Through simulation experiments, the paper studies and predicts the response to the takeover of silent automatic vehicle faults, and puts forward the corresponding prediction model. The research makes sense.

In the previous comment reply, the author has given a comprehensive explanation and improvement to the experimental process, the structure of the paper and the result statistics. I suggest that the manuscript give a supplement and explanation to the manuscript.

Reviewer #2: (No Response)

7. PLOS authors have the option to publish the peer review history of their article (what does this mean?). If published, this will include your full peer review and any attached files.

Reviewer #1: No

Reviewer #2: No

---

## [Editor Report · Acceptance letter]

16 Nov 2020

PONE-D-20-24803R1

Predicting takeover response to silent automated vehicle failures

Dear Dr. Mole:

I'm pleased to inform you that your manuscript has been deemed suitable for publication in PLOS ONE. Congratulations! Your manuscript is now with our production department.

Kind regards,

on behalf of

Dr. Feng Chen

Academic Editor

PLOS ONE